# Enrichment of Pistachio Shell with Olive Mill Waste or *Lathyrus clymenum* Pericarp Mixtures via Solid State Fermentation with *Pleurotus ostreatus*

Christos Eliopoulos [1,2], Giorgos Markou [1], Alexandra Kremmyda [3], Serkos A. Haroutounian [2] and Dimitrios Arapoglou [1,*]

1 Institute of Technology of Agricultural Products, HAO-DEMETER, Sof. Venizelou 1, 14123 Athens, Greece; chris_eliopoulos@hotmail.com (C.E.); markougior@gmail.com (G.M.)
2 Laboratory Nutritional Physiology & Feeding, Department of Animal Science, Agricultural University of Athens, Iera Odos 75, 11855 Athens, Greece; sehar@aua.gr
3 Division of Food, Nutrition and Dietetics, School of Biosciences, Sutton Bonington Campus, University of Nottingham, Loughborough LE12 5RD, UK; stxak39@nottingham.ac.uk
* Correspondence: darapoglou@itap.com.gr; Tel.: +30-210-284-5940

**Abstract:** The study herein concerns the application of the solid-state fermentation (SSF) bioprocess of agro-industrial wastes as a means to improve their nutritional composition, targeting their utilization as proteinaceous animal feed. The fermentation outcome resulted from the mixtures of Olive Mill Stone Waste (OMSW) with Pistachio Shell (PS) and PS with *Lathyrus clymenum* pericarp (LP) at various proportions via SSF initiated by *P. ostreatus.* The addition of 20% *w/w* of LPs to PS recorded the highest crude protein content (%) increase of 33.87% while concerning cellulose content, 50% *w/w* addition presented the highest value (37.68%). Concerning lignin presence, PS and its additions to OMSW recorded a reduction, the ratio of 100% *w/w* of PS was found to be decreased by 14.22% whereas, 20% *w/w* of LP additions to PS displayed an increment of 38.25%. Regarding *β*-glucans content, the mixture of 50% *w/w* of LP to PS recorded the highest value (5.19%) while 100% *w/w* of PS presented a vast increment exceeding 480-folds. The OMSW, PS and LP mixtures revealed their potential as supplements in animals' diets after their nutritional upgrade through SSF. Such studies highlight the contribution to the confrontation of the unavailability of proteinaceous animal feed in the terms of a circular economy.

**Keywords:** sustainable management; *Pleurotus ostreatus*; crude protein enrichment; waste valorization; olive mill stone waste; pistachio shell; *Lathyrus clymenum* pericarp



## 1. Introduction

By 2050, the world population is expected to show a sharp increase. Global food production has to be increased by 60% in order to fulfill these augmented nutritional needs by simultaneously contributing to a higher production of agro-food wastes [1]. In recent decades, the food and agriculture industries recorded a significant development by resulting in their intense industrialization. During 2018, fishing, forestry and agriculture sectors produced 21 Mtons of wastes in the European Union by their disposal constituting a major environmental issue [1]. Their disposal to land fields is associated with increased environmental pollution and negative impacts on human and animal health [2,3]. The European Union has focused on the implementation of the circular economy by applying wastes' biotransformation targeting to their depletion, in order to avoid their burning or landfilling [1]. There has been a great interest to confront this situation either by their proper management or by their reuse [4]. Annually, residues originated from the agro-industrial sector are estimated to reach five million metric tons of biomass which can be classified as lignocellulosic [2]. They are mainly composed of polysaccharides such as

Cellulose, Hemicellulose, Lignin, and additionally they are rich in nutrients, minerals sugars and proteins. Due to their nutritional composition, they can be characterized as raw materials which can be easily assimilated by the microorganisms. Microorganisms' growth depends on the existence of carbon, nutrients and moisture, consequently the aforementioned raw materials can form an ideal substrate for their development [2,3,5]. Under this point of view, numerous research endeavors have been initiated, aiming to the upgrade and exploitation of these raw materials indicating their nutritional perspectives for the production of high value-added products via solid-state fermentation (SSF) process [6]. SSF is called the bioprocess which is applied in the absence of water but with adequate levels of moisture by enhancing/supporting microorganism's growth [7].

Usually, filamentous fungi are employed for SSF processes due to their ability on growing and colonizing biomass with lignocellulosic nature. *Pleurotus ostreatus* is classified as an edible white rot fungus, well known as a potent mean for lignin degradation by secreting extracellular hydrolytic and oxidative enzymes. Lignocellulolytic enzymes include ligninolytic (peroxidases and oxidases) and cellulolytic enzymes (cellulases and hemicellulases) [8]. Previous studies concerning ruminants feeding with fermented biomass by white rot fungus, presented beneficial properties to them, such as increased digestibility, improved feed efficiency ratio and welfare [9].

Mediterranean basin countries such as Greece, Spain, Italy, etc. are the main producers of olive oil. Olive oil production is being performed by using a two or three-phase extraction systems where a lot of wastes are derived. Olive press cake and a liquid effluent with dark color which is referred as wastewater, are the main wastes derived from a three phase system. This specific system, has the drawback of using vast amounts of water (up to 50 L/100 kg olive paste), resulting in an annual wastewater production of 30 million m$^3$. These wastes are mainly disposed in soils and rivers and are considered as toxic due to their high concentration in polyphenolic content, contributing to the environmental pollution [10,11].

Pistachio (*Pistacia vera* L.) is commonly cultivated in arid regions of Central and West Asia and is classified to the Anacardiaceae family. Pistachio fruit is mainly consumed as food. For 2016, pistachio's production reached almost 1,057,566 tons worldwide, where USA and Iran produced about 68% forming the predominant producers, followed by Turkey (16%), China (7.5%) and Syria (5%) [12]. Furthermore, for 2018/2019 its annual global production was estimated at 780,306 tons in shell basis, consequencing to the production of significant amounts of Pistachio Shell (PS) [13]. PS is one of the main byproducts of the respective cultivation which is produced in large volumes. Its weight represents the 47% of the total nut weight and is usually burned or discarded as solid waste in land fields [13,14]. It is extremely harmful for the environment and for the human health due to the methane production, so its disposal forms a serious environmental issue [14]. PS is a promising and economical residue by indicating the potential for bioconversion under SSF process, into biopolymers, biofuels, etc. reaping thus its physicochemical and nutritional benefits [13].

Legumes represent a large category of cultivated plants including *Lathyrus* species. They are well known for their nutritional value especially for their seeds' high protein content, minerals and vitamin concentration [15]. *Lathyrus* species constitute an intriguing case hence their cultivation is capable to improve soil physical conditions, diminish diseases and weed infestations and finally reduce production costs. [16]. *Lathyrus clymenum* is an annual species which is classified into the Fabaceae family. It is originated from the Mediterranean basin countries and is still cultivated in Aegean islands such as Thera (Santorini), Anafi and Karpathos. Its nutritional composition follows the same pattern as the latter's *Lathyrus* species. To date, it is still cultivated and consumed as a food and feed [17]. *Lathyrus clymenum* pericarp (LP) is considered the main by-product of the respective cultivation. It is composed of a complex of polysaccharides (Cellulose, Hemicellulose, Lignin) enhancing ruminants with energy, but also is considered as a low quality feedstock, regarding its low protein content and the lack of some nutritional factors such as digestibility, palatability and bulkiness.

A common feature of all previous residues is their poor nutritional value, which forms the major inhibitory factor for their valorization in livestock feeding. This fact generates the necessity for the development of new methods in order to achieve their nutritional upgrade by making them suitable for exploitation in animal feeding [18]. As we have already pointed out, a possible potent means in order to deal with low protein content as well as fiber substances presence requires the utilization of white-rot fungi such as *P. ostreatus*, which has the potential to upgrade their protein content and degrade their crude fiber substances. Considering the environmental aspects, as well as the lack of proteinaceous feedstuffs, it is imperative to find new alternative sources for exploitation in the animal sector. In this respect, the valorization of agro-industrial residues is considered as an intriguing case contributing thus to the circular economy as well as to the production of new eco-friendly feedstuffs.

The goal of this study is the exploitation of PS with or without adding OMSW or LP at various ratios, into a novel and nutritional upgraded proteinaceous animal feed, through a solid-state fermentation process initiated by *P. ostreatus*.

## 2. Materials and Methods

### 2.1. Materials and Microorganisms

OMSW, PS and LP were used as substrates for the growth of *P. ostreatus*. For the achievement of the necessary moisture content which is essential for SSF performance and *P. ostreatus'* development, in order to colonize completely the examined substrates, tap water was added to all substrates and renewed every day [19–21]. The initial dry substrate was milled to a particle size of 2mm, which allowed oxygen diffusion and facilitated the fungal colonization. The hydrated samples were weighed and mixed to a final weight of 300 g by mixing OMSW with 10% *w/w* and 20% *w/w* proportions of PS, as well as PS with 20% *w/w* and 50% *w/w* proportions of LP and, finally, PS was used separately. The fermentation process was performed into closed glass test vessels of 750 mL volume. In specific, 300g of each examined substrate were placed into test vessels and then were sterilized by heating at 121 °C for 15min. Subsequently, 3% *w/w* of inoculum was added in the center of each substrate for the inoculation process. Incubation was carried out in a bioclimatic chamber with a stable temperature at 25 °C in the absence of light. The inoculum which was used in our study is a commercial commodity, *Pleurotus ostreatus* White 2000 P67 LOTTO 1551 MN 01827 (Fungi SEM, La Rioja, Spain) strain, in which the fungi have colonized barley seeds and throughout the experiment was stored at 4 °C.

### 2.2. Inoculation and Solid-State Fermentation

Inoculation was performed in a vertical laminar flow chamber. Inoculum of *P. ostreatus* strain was used in a ratio of 3% *w/w* and subsequently, was added in the center of each substrate and transferred into an incubator at 25 °C. The total incubation time was 21 days and for analysis, samples were taken at Day 0, Day 11 and Day 21. Every sample was prepared and studied in triplicates.

### 2.3. Analytical Methods

Moisture, crude protein content and ashes were determined according to the AOAC methods [22]. Total soluble sugars were determined according to Dubois et al., [23] and reducing sugars were determined according to Miller [24]. Sugars' assessments were performed on aqueous extracts of the respective samples. Crude fiber substance content was determined in accordance with the AOAC method [22]. Cellulose and lignin presence was evaluated according to the acid-detergent fiber (ADF) method [22]. Finally, $\beta$-glucans were assessed using the MEGAZYME enzymatic assay kit ($\beta$-Glucan Assay Kit Yeast & Mushroom, Megazyme Product code: K-YBGL, Bray, Ireland). The results of the aforementioned analyzes are expressed as g/100 g of dry weight.

### 2.4. Statistical Analysis

All analyses were performed in triplicates and the results were expressed in means ± standard deviation (±S.D.). Data normality was assessed using the Kolmogorov–Smirnov and Shapiro–Wilk test. The differences between the groups were analyzed by paired *t*-test ($p \leq 0.05$ was considered significant). To all proportions of both substrates, a statistical analysis was performed between Day 0 and Day 21.

## 3. Results

Figure 1 illustrates the gradual mycelium growth for all the examined substrates at the baseline of fermentation process (Day 0) until the end of fermentation (Day 21). Table 1 presents the results for the additions of 10% *w/w* and 20% *w/w* of PS to OMSW, the ratio of 100% *w/w* of PS as well as 20% *w/w* and 50% *w/w* of LPs to PS for the physicochemical parameters of Moisture content, Total and Reducing Soluble Sugars concentration and Ash presence of the examined substrates at Day 0 and Day 21. The moisture content for 10% *w/w* and 20% *w/w* additions of PS to OMSW ranged from 41.81% to 47.96% and 48.68% to 51.32%, respectively, between Days 0 and 21. The proportion of 100% *w/w* of PS was augmented by 13.70% without significance ($p \leq 0.05$), whereas, for 50% *w/w* and 20% *w/w* of LPs additions, there was an increase by 2.30% and 12.76%, respectively, without significance ($p \leq 0.05$) was observed between Days 0 and 21. The concentration of Total Soluble Sugars (TSS) underwent a 2-fold statistically increase for the 10% *w/w* addition of PS whereas, 20% *w/w* addition was decreased by 24.02% without significance ($p \leq 0.05$). The substrate of 100% *w/w* of PS displayed an increase without significance ($p \leq 0.05$) by 43.42% during Day 0 and Day 21. LPs addition of 50% *w/w* was significant decreased exceeding 2-folds ($p \leq 0.05$) along with 20% *w/w* addition, by presenting similar results, hence TSS content was diminished by 29.79 % between Day 0 and 21, respectively.

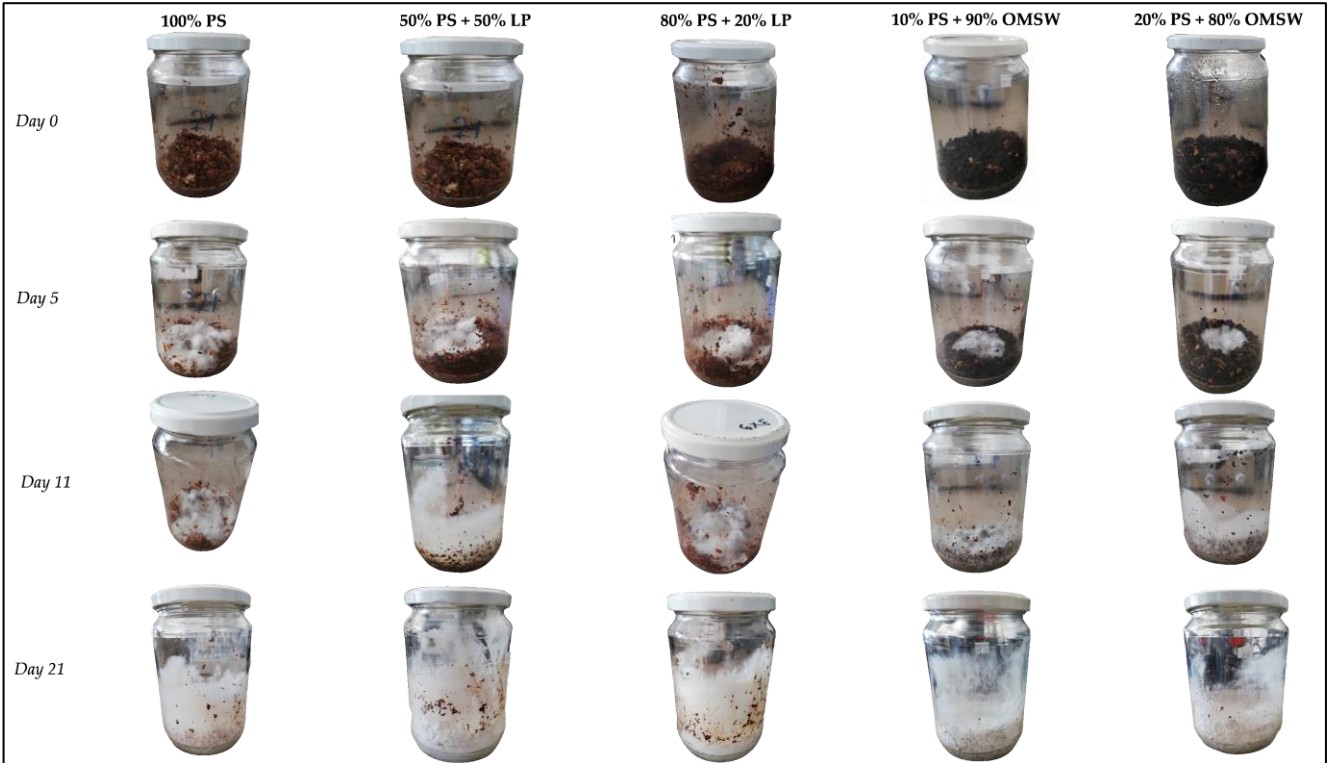

**Figure 1.** Representative images of mycelial growth for PS and its additions of 10% *w/w*, 20% *w/w*, 50% *w/w* and 20% *w/w* to OMSW and LP, respectively, at the beginning (Day 0) and at end of fermentation (Day 21).

**Table 1.** Assessment of Physicochemical Parameters for PS and its additions to OMSW and LPs.

| Ratios (% *w/w*) | Moisture (%) | | TSS (%) | | RSS (%) | | Ash (%) | |
|---|---|---|---|---|---|---|---|---|
| | Day 0 | Day 21 | Day 0 | Day 21 | Day 0 | Day 21 | Day 0 | Day 21 |
| OMSW + PS | | | | | | | | |
| 90–10 | 41.81 ± 1.11 | 47.96 ± 0.95 | 6.64 ± 0.19 [a1] | 13.65 ± 0.29 [a2] | 0.66 ± 0.02 [b1] | 2.04 ± 0.17 [b2] | 3.63 ± 0.18 | 3.63 ± 0.19 |
| 80–20 | 48.68 ± 0.02 | 51.32 ± 3.03 | 10.74 ± 0.27 | 8.16 ± 0.96 | 0.77 ± 0.05 | 0.75 ± 0.15 | 3.59 ± 0.19 | 3.68 ± 0.37 |
| PS | | | | | | | | |
| 100 | 51.97 ± 0.60 | 59.09 ± 5.71 | 13.47 ± 0.84 | 19.32 ± 0.74 | 1.56 ± 0.09 [c1] | 0.38 ± 0.02 [c2] | 1.42 ± 0.02 | 1.57 ± 0.06 |
| PS + LP | | | | | | | | |
| 50–50 | 64.51 ± 0.12 | 66.00 ± 2.98 | 22.48 ± 1.40 [d1] | 10.90 ± 1.58 [d2] | 1.91 ± 0.28 | 1.78 ± 0.18 | 1.56 ± 0.22 | 1.81 ± 0.13 |
| 80–20 | 55.24 ± 1.14 | 62.29 ± 9.12 | 27.92 ± 1.67 | 19.60 ± 0.24 | 2.54 ± 0.98 [e1] | 1.64 ± 0.01 [e2] | 0.93 ± 0.13 [f1] | 1.62 ± 0.10 [f2] |

TSS: Total Soluble Sugars, RSS: Reducing Soluble Sugars. Statistical analysis was performed for each examined parameter and substrate individually between Day 0 and Day 21. Different superscripts between the same letters indicate statistical significance ($p \leq 0.05$).

For 20% *w/w* addition of PS to OMSW Reducing Soluble Sugars (RSS) content was practically unaffected since a slight decrease was observed (2.59%), along with 100% *w/w* ratio of PS which depicted a statistically significant decrease of 75.64% ($p \leq 0.05$) between Days 0 and 21. The ratio of 10% *w/w* addition of PS to OMSW fluctuated from 0.66% to 2.04% presenting the highest statistically significant increase ($p \leq 0.05$) from Day 0 to Day 21. RSS concentration was in line with the respective results concerning LPs additions to PS. In particular, for 50% *w/w* addition, TSS content ranged from 1.91% to 1.78% while a notable reduction for 20% *w/w* addition was observed hence RSS content was significantly reduced by 35.43% ($p \leq 0.05$).

Concerning PS to OMSW addition by 10% *w/w*, ash presence was not practically varied between Day 0 and Day 21 hence its content remained unaffected whereas, as for PS and its addition by 20% *w/w* a slight increase of 10.56% and 2.50%, respectively, was displayed. LPs additions to PS presented an increment concerning ash content. In specific, 50% *w/w* addition recorded an increase of 16.02% while 20% *w/w* addition resulted to a statistically significant increase ranging from 0.93% to 1.62% ($p \leq 0.05$).

According to Figure 2, all proportions recorded an increase in crude protein content (%) during the fermentation period. In particular, 20% *w/w* proportion of PS ranged from 6.97% to 7.27% whereas, the PS proportion of 100% *w/w* as well as its addition by 10% *w/w* to OMSW recorded a significant increase of 24.87% and 25.58% respectively.

As shown to Figure 3 all proportions of mixed PS&LPs as well as 100% *w/w* of PS proportion, resulted to augmented crude protein content (%) between Day 0 and Day 21. LPs addition of 50% *w/w* demonstrated an increase of 20.99%, the 100% *w/w* PS substrate presented a significant increase of 24.87% and the addition of 20% *w/w* initiated the highest crude protein content (%) by recording a significant increment of 33.87%.

The Crude Fiber Substances content was found to be practically unaffected for all ratios studied (Figure 4) during Day 0 and Day 21. For 10% *w/w* addition of PS to OMSW a slight decrease was observed (0.06%), whereas 20% *w/w* addition of PS and 100% *w/w* of PS ratio were found to be slightly increased of 0.95% and 1.92%.

Figure 5 tabulates the Crude Fiber Substances content for all examined ratios of PS and LPs mixtures as well as for 100% *w/w* PS ratio. More particularly, 20% *w/w* addition was decreased ranging from 48.14% to 46.43%, 50% *w/w* addition displayed a significant increment of 6.96% ($p \leq 0.05$) and finally, 100% *w/w* of PS also displayed a slight increase of 0.95% between Day 0 and Day 21.

According to Figure 6, Cellulose content was augmented for all the examined proportions. In specific,100% *w/w* of PS and its additions of 10% *w/w*, 20% *w/w* were increased by 12.97%, 12.96%, and 17.04%, respectively, by 20% *w/w* addition displaying the highest statistically significant increase ($p \leq 0.05$).

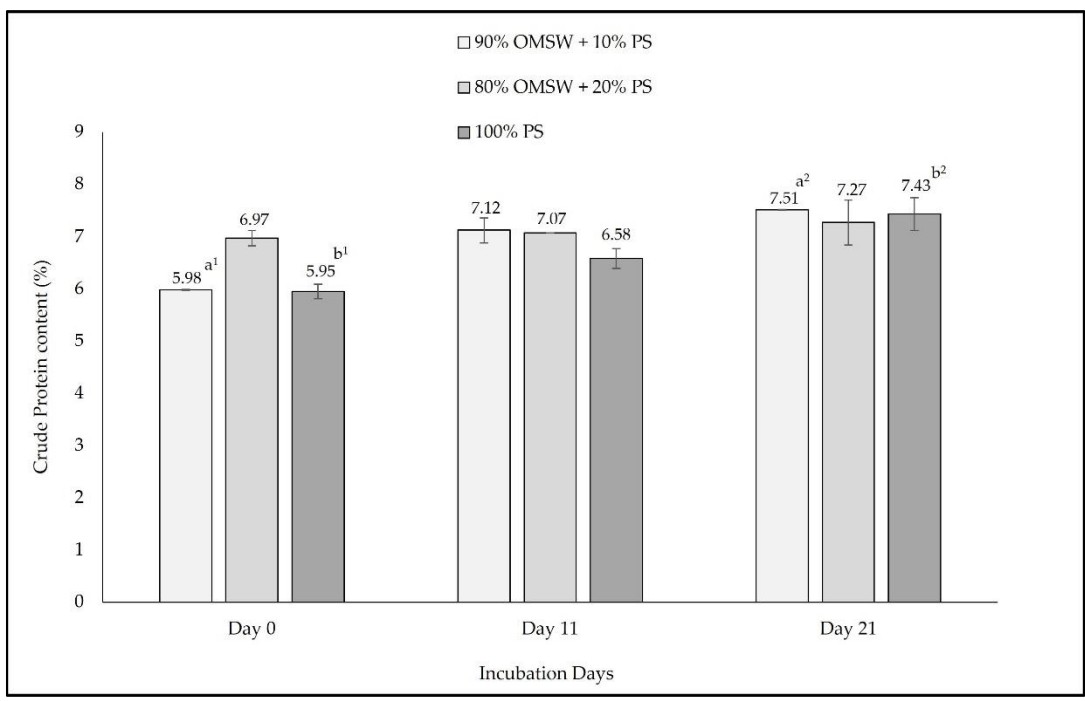

**Figure 2.** Evaluation of crude protein content (%) for PS and its mixtures with OMSW. Statistical analysis was performed for each substrate individually between Day 0 and Day 21. Different superscripts between the same letters indicate statistical significance ($p \leq 0.05$).

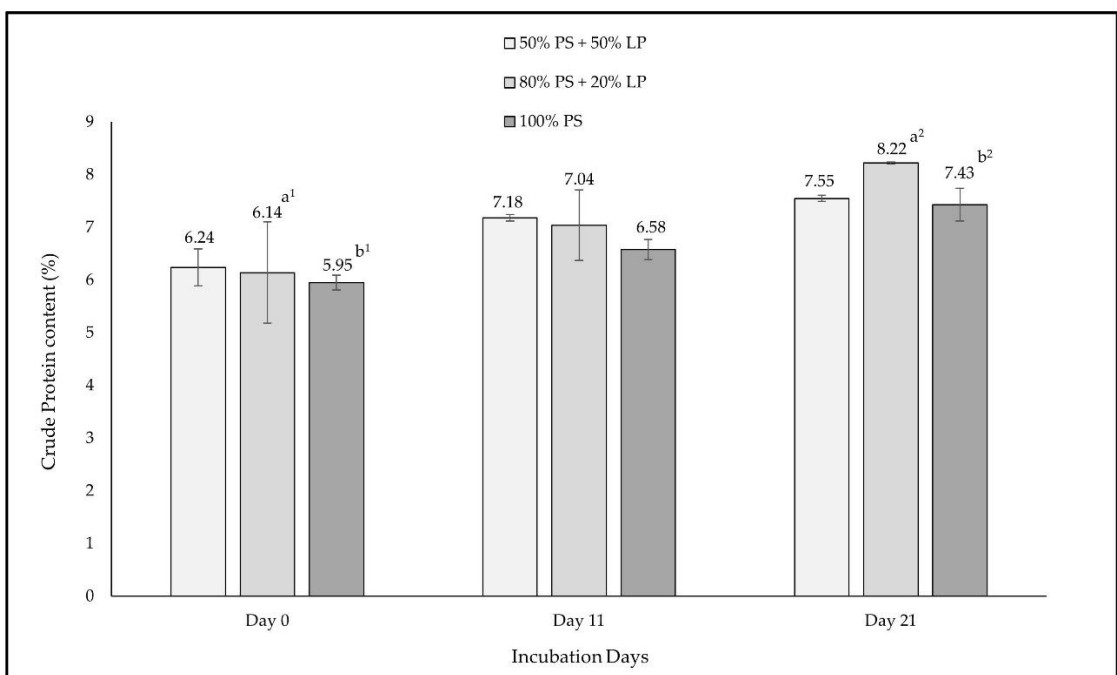

**Figure 3.** Evaluation of crude protein content (%) for PS and its mixtures with LPs. Statistical analysis was performed for each substrate individually between Day 0 and Day 21. Different superscripts between the same letters indicate statistical significance ($p \leq 0.05$).

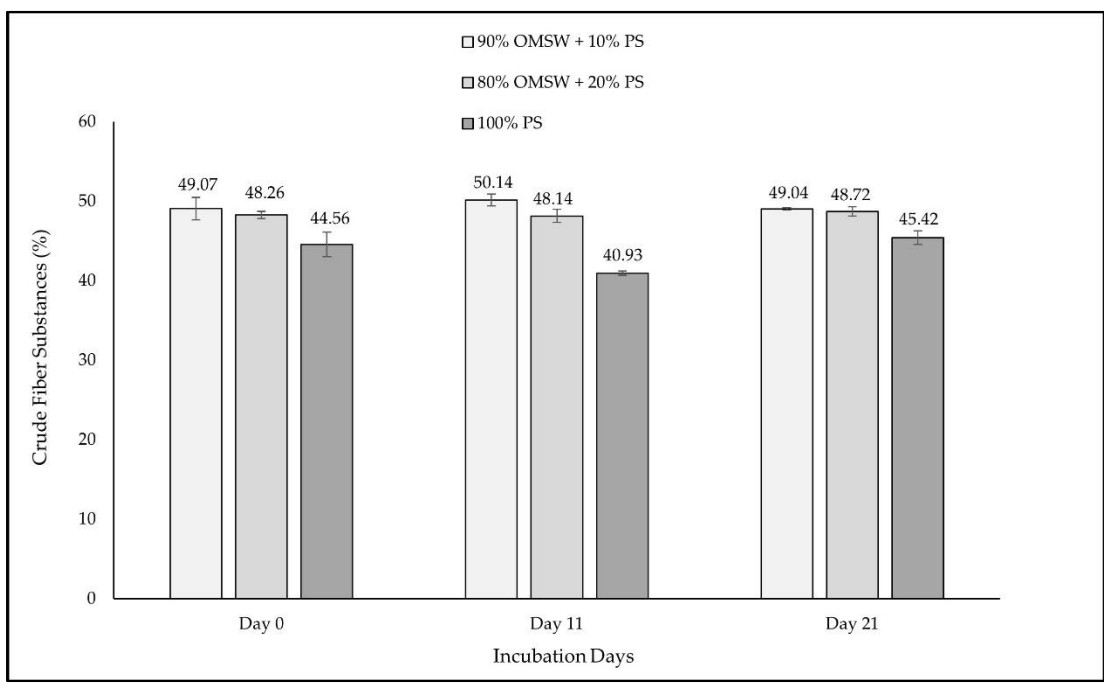

**Figure 4.** Assessment of Crude Fiber Substances content for PS and its mixtures with OMSW. Statistical analysis was performed for each substrate individually between Day 0 and Day 21.

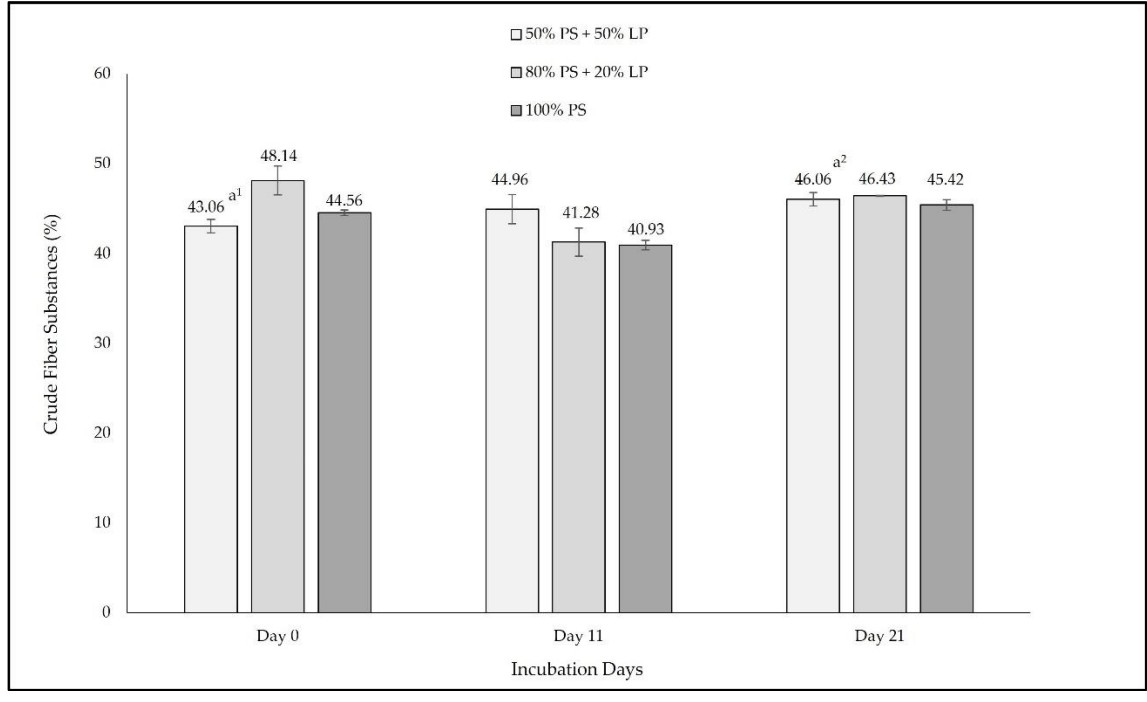

**Figure 5.** Assessment of Crude Fiber Substances content for PS and its mixtures with LPs. Statistical analysis was performed for each substrate individually between Day 0 and Day 21. Different superscripts between the same letters indicate statistical significance ($p \leq 0.05$).

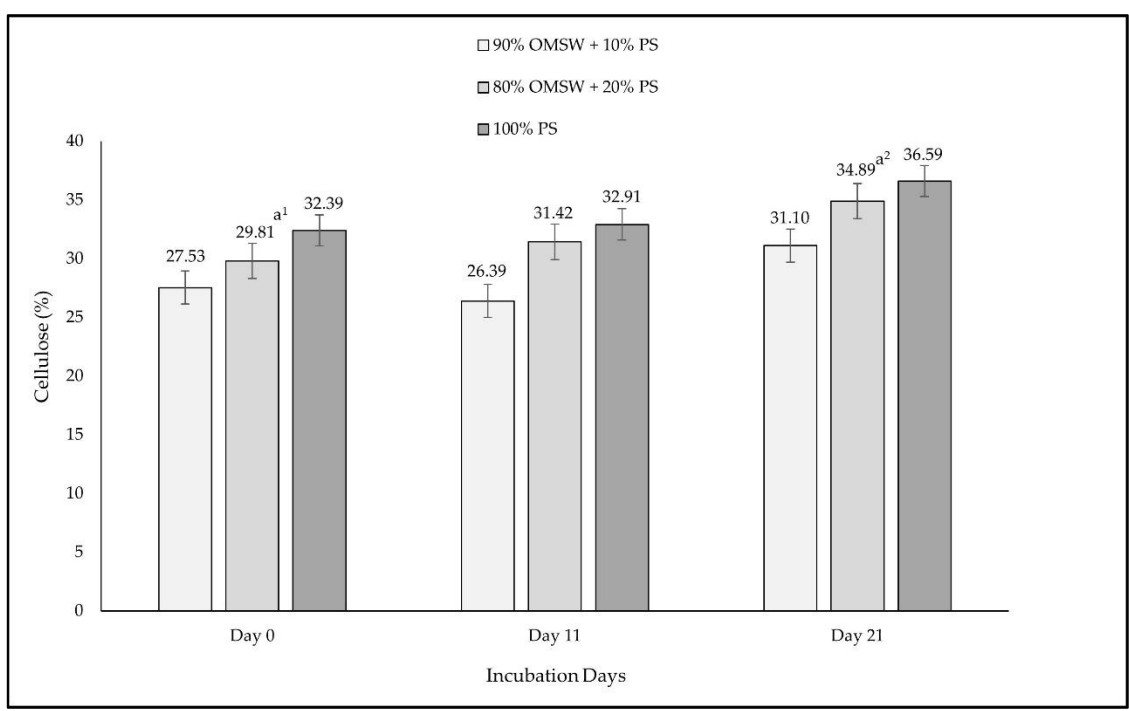

**Figure 6.** Evaluation of Cellulose content for PS and its mixtures with OMSW. Statistical analysis was performed for each substrate individually between Day 0 and Day 21. Different superscripts between the same letters indicate statistical significance ($p \leq 0.05$).

A similar pattern was observed regarding Cellulose content for all the examined ratios of LPs additions as well as 100% $w/w$ of PS. As shown in Figure 7, all examined ratios presented an increment to their cellulose content. Concerning 10% $w/w$ addition, cellulose content fluctuated from 35.26% to 37.68%, 20% $w/w$ addition recorded a statistically significant increase of 6.92% while 100% $w/w$ proportion ranged from 32.39% to 36.59%.

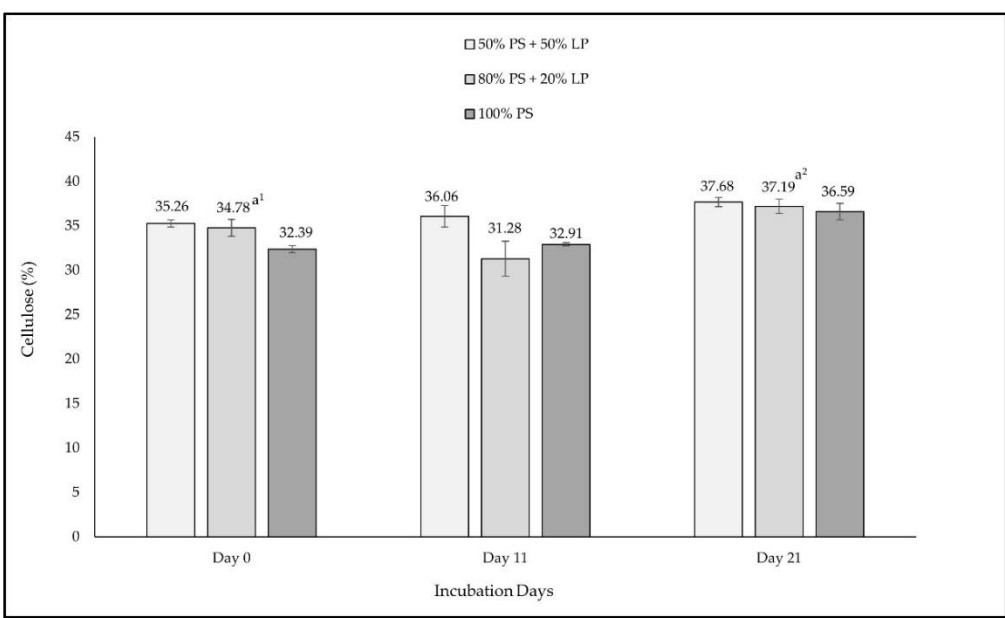

**Figure 7.** Evaluation of Cellulose content for PS and its mixtures with LPs Statistical analysis was performed for each substrate individually between Day 0 and Day 21. Different superscripts between the same letters indicate statistical significance ($p \leq 0.05$).

Lignin concentration (Figure 8) was found to be reduced for all the examined ratios concerning 100% *w/w* of Ps and its additions by 10% *w/w* and 20% *w/w* of PS to OMSW, by the substrate of 20% *w/w* resulting in a statistically significant decrease, while 100% *w/w* ratio of PS displayed the highest decrease of 14.22% without significance.

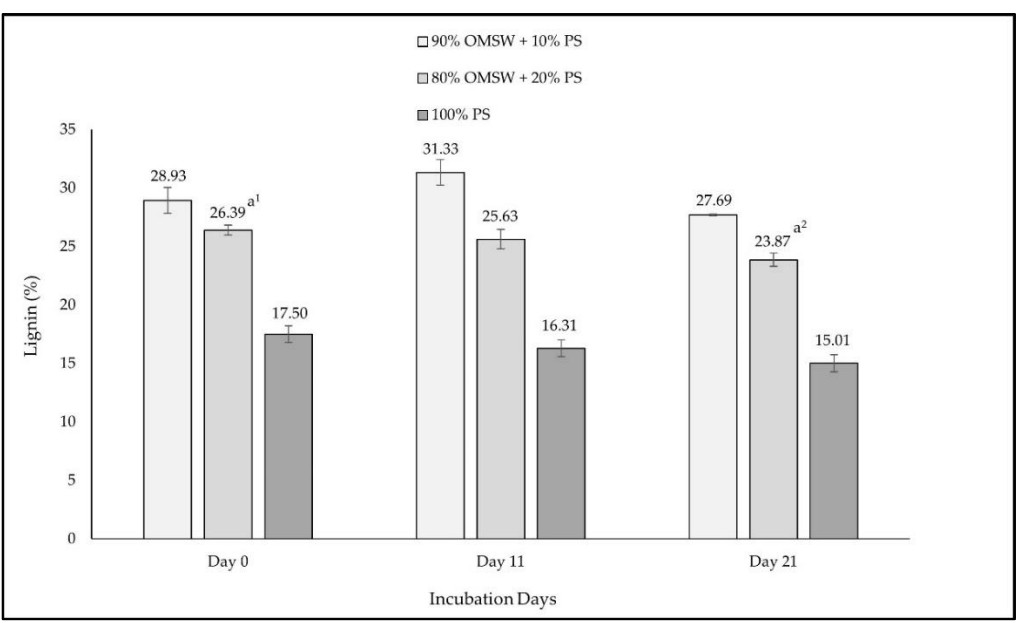

**Figure 8.** Assessment of Lignin content for PS and its mixtures with OMSW. Statistical analysis was performed for each substrate individually between Day 0 and Day 21. Different superscripts between the same letters indicate statistical significance ($p \leq 0.05$).

The lignin content (Figure 9) of the studied ratios of LPs additions to PS was increased, whereas the substrate of 100% *w/w* of PS was reduced. In particular, PS additions of 10% *w/w* and 20% *w/w* presented an increment of 11.46% and 38.25% with the latter displaying the highest increase. Finally, concerning 100% *w/w* of PS ratio displayed the highest decrease of 14.22% without significance.

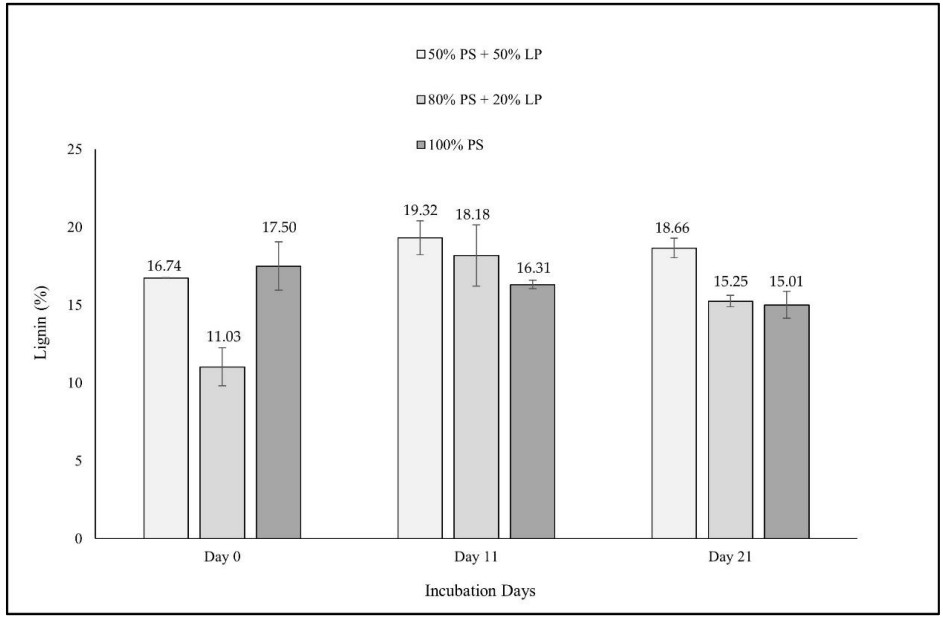

**Figure 9.** Assessment of Lignin content for PS and its mixtures with LPs. Statistical analysis was performed for each substrate individually between Day 0 and Day 21.

According to Figure 10, $\beta$-glucans content for all the examined proportions was found to be augmented between Days 0 and 21 by the substrate of 100% $w/w$ of PS resulting in the highest statistically significant increase.

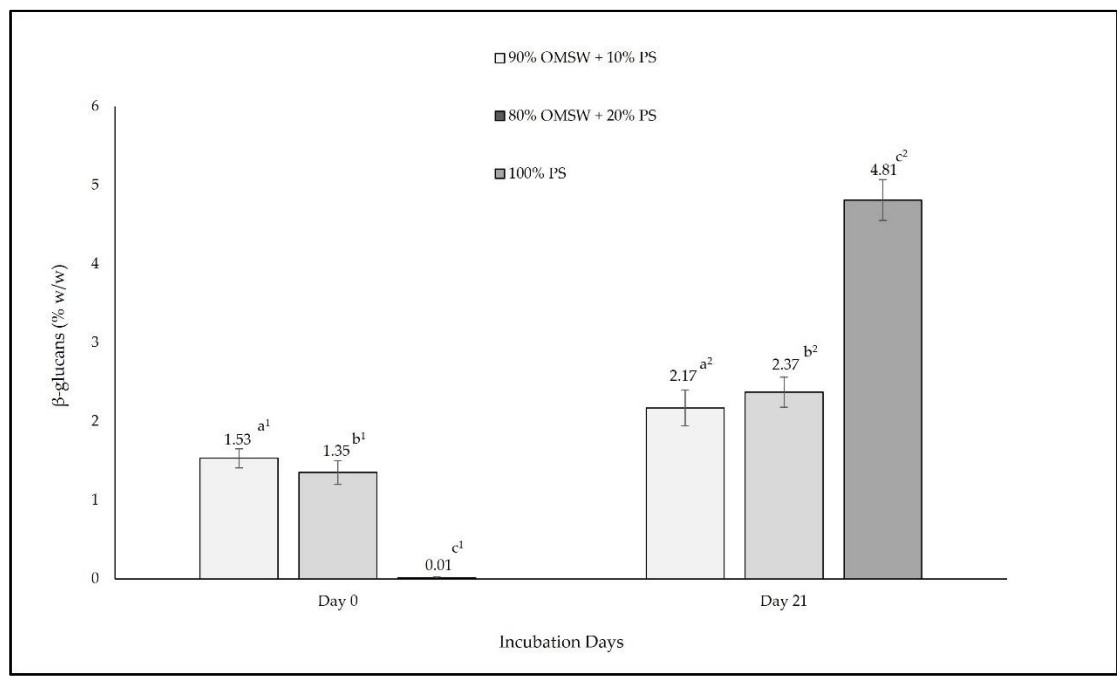

**Figure 10.** Evaluation of $\beta$-glucans content for PS and its mixtures with OMSW. Statistical analysis was performed for each substrate individually between Day 0 and Day 21. Different superscripts between the same letters indicate statistical significance ($p \leq 0.05$).

$\beta$-Glucans content for LPs additions was in line with the respective results for PS additions to OMSW (Figure 11). In specific, 50% $w/w$ addition of LPs to PS ranged from 3.88% to 5.19%, LPs addition of 20% $w/w$ exceeded a 2-fold increase while the substrate of 100% $w/w$ of PS recorded the highest statistically significant increase which was consequently the highest level of observed $\beta$-glucans content.

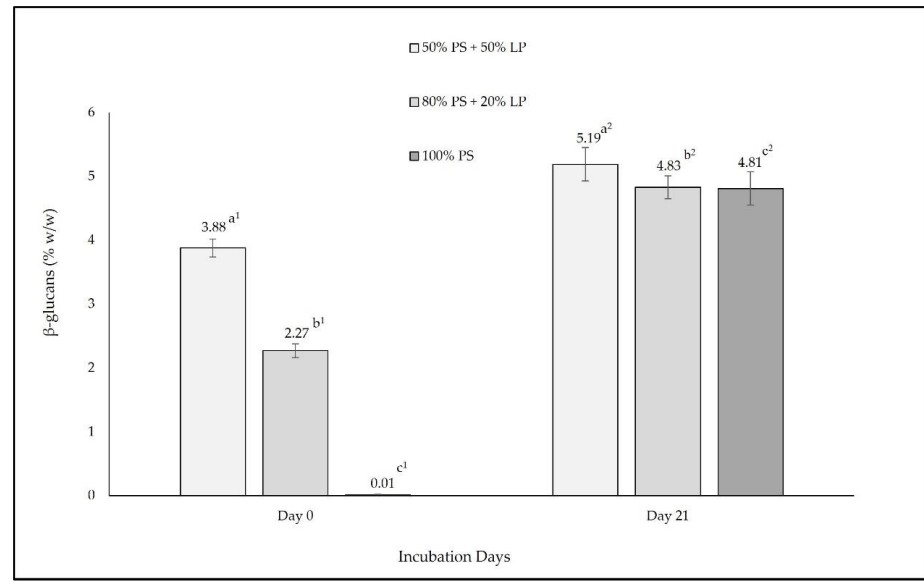

**Figure 11.** Evaluation of $\beta$-glucans content for PS and its mixtures with LPs. Statistical analysis was performed for each substrate individually between Day 0 and Day 21. Different superscripts between the same letters indicate statistical significance ($p \leq 0.05$).

## 4. Discussion

The main objective of this work is the valorization of PS, PS and LPs mixtures with OMSW, as substrates applying the SSF process initiated by *P. ostreatus*. The final goal of the study is the evaluation of the fermentation outcome as proteinaceous used in animal feed.

Moisture content impacts nutrient availability and oxygen transfer, since it forms a critical factor for the SSF process [17,21]. In this respect, if the moisture content is lower than the essential levels, nutrients' solubility and accessibility are limited consequencing in poor fungal growth, since fungi's nutrient uptake is not required [19,20]. On the other hand, high moisture content contributes to fungal growth, nutrients transportation and enzyme activities but also prevents oxygen's transfer, as well as promotes the risk for contamination initiated mainly by filamentous fungi [20]. The optimum moisture content is affected by many factors in closed systems, such as the lignocellulosic nature of the examined substrates, which is very important for the achievement of the desired moisture content. According to Nagel et al. [25], water balance in SSF can be affected by four major factors including the required water for starch hydrolysis, metabolic water production, intracellular water's intake during biomass production and finally water evaporation as an effect from the production of metabolic heat. These four factors could possibly justify an increase in moisture content in the closed systems.

Fan et al. [19] reported that the ratio of 60% is considered as the most essential moisture content for fungal growth. These findings are in accordance with our results since in our experiments, moisture content was preserved at 60% for the mixtures of LPs additions to PS compared to OMSW which reached almost 50%. This reduction is probably a result of the closed system and of the lignocellulosic structure of the substrates without however provoking any significant issues to SSF process.

The most important examined physicochemical parameter of SSF process is substrates' upgraded crude protein content. Crude proteins' content bioconversion is indicative of the method's efficiency and can be affected by temperature, oxygen and nutrient availability. After the colonization of *P. ostreatus* crude proteins' content was increased. This fact could be probably attributed to fungal biomass accumulation [26,27]. According to Akinfemi et al. [28] and Oseni and Akindahunsi [29], another fact that could potentially contribute to the augmented crude protein content, is the secretion of some proteinaceous extra cellular enzymes by degrading the by-products during fermentation, which are metabolized. Akinfemi et al. [28] also reported that the uptake of nitrogen excess through aerobic fermentation could possibly be one more reason associated with crude proteins' increment. Nasehi et al. [30] reported that after SSF performance initiated by *P. florida* on some agricultural by-products, crude protein content was found to be significantly increased. Additionally, Parani and Eyini [31] examined the biodegradation abilities of various fungi including *P. ostreatus* at coffee pulp. At the end of the SSF process, the authors mentioned an increment in crude protein content. Accordingly, in another study where *P. ostreatus* was employed in order to facilitate SSF on native rice husk crude proteins revealed an increase during SSF procedure [32]. Eliopoulos et al. [33] examined BSG's bioconversion applying SSF process initiated by *P. ostreatus.* The authors stated that at the end of the procedure the crude protein content was significantly increased. Finally, an investigation concerning the bioconversion of olive cake through SSF was carried out by Chebaibi et al. [34]. Some filamentous fungi were employed in order to upgrade olive cake's nutritional content by crude proteins' concentration displaying a significant increment. Results obtained herein are in the line with the previous literature reports, hence crude protein content was increased to all studied proportions. Addition of 10% $w/w$ PS to OMSW and 20% $w/w$ LPs to PS were found to be the most indicative by displaying a significant increase of 25.58% and 33.87%, respectively, at the end of fermentation (Day 21). It must be noted that the proportion of 100% $w/w$ of PS displayed an almost similar significant increase of 24.87% with the addition of 10% $w/w$ PS to OMSW.

Fungi and especially basidiomycetes are considered the most efficient degraders of lignocellulosic substrates. The fungus has the ability to exploit the substrates' nutrients in

order to produce new organic substances, which can be transported via fungus to the entire substrates' biomass [35]. White Rot Fungi such as *P. ostreatus* have two types of extracellular enzymatic systems: the hydrolytic system, where hydrolases are secreted and are responsible for polysaccharide degradation, and a unique oxidative and extracellular ligninolytic system, which degrades lignin and generates phenyl rings [36]. Laccase, Lignin Peroxidase and Manganese Peroxidases are considered to be the most indicative extracellular enzymes for lignin degradation. Furthermore, it must be noted that *Pleurotus* spp displays the ability to degrade more lignin than cellulose [37]. The oxidative activity of the *Pleurotus* spp, as well as their ligninolytic enzymes, make the latter species capable to reduce the phytotoxicity wastes which are associated with some toxic substances like phenols and tannins. Furthermore, the exploitation of food industry by-products as substrates for the cultivation of edible mushrooms constitutes an intriguing case, since it contributes to the reduction of the environmental impact as well as in the production of mushrooms with amplified nutritional content [37,38].

Herein, cellulose content displayed an increase for all studied substrates, whilst the presence of lignin was found to be diminished for PS and its additions to OMSW, as compared to LPs additions to PS, where lignin was increased during the fermentation process. For the examined substrates of LPs additions to PS, cellulose and lignin increment can be probably rationalized due to the different fiber fractions content of the initial raw materials. Another reason could be substrates' ability to provide the fungus with nutrients probably N, P, and K, resulting thus in a more extensive degradation [9]. Furthermore, lignin increment could possibly be correlated with the reduction of hemicellulose [27]. It must be noted that *Pleurotus* spp. are considered as valuable microorganisms suitable for the valorization and nutritional enhancement of agricultural by-products. Considering our results, PS additions to OMSW recorded a reduction concerning lignin content due to the efficiency of *P. ostreatus* to act as lignin degradation agent by secreting the aforementioned specific extracellular enzymes. This fact secures the ability to improve the substrates' digestibility, making them suitable for feedstuffs [37].

It must be reported that the consumption of cellulose contributes to the feeling of satiety in animals. Hanczakowska et al. [39] reported that the addition of cellulose on piglets' diet has beneficial impacts on their health and rearing performance since cellulose consumption contributed to the reduction of diarrhea incidences and mortality rate, whereas Pascoal et al. [40] also stated that cellulose consumption revealed positive effects in some immunological parameters. Another advantage of feedstuffs with high cellulose content is associated with the alteration of the stomach and large intestine development, targeting small intestinal barrier function improvement, to the reduction of pathogens proliferation, and finally to the fecal consistency improvement in post-weaning pigs [41]. The fluctuation of cellulose and lignin content was reflected in the crude fiber substances concentration.

Regarding PS and its additions to OMSW, crude fiber substances were practically unaffected hence a slight increase was observed. The same pattern was displayed for the 50% *w/w* of LPs addition while 20% *w/w* was slightly reduced.

Ash content was found to be augmented for all the examined ratios at Day 21 except from 10% *w/w* addition of PS to OMSW where it was unaffected. This increase could be possibly associated with *P. ostreatus* which contributes to the fermentation process [29,42]. Rajesh et al., [43] reported that the observed increase of ash content may be associated with organic matter's depletion which is performed during SSF, whereas Okpako et al. [44] highlighted the importance of ash content increment since ash is considered as an indicator of minerals content determination. Finally, the authors stated that increased ash levels after the fermentation process could BE useful in animal feeds.

As for TSS and RSS content, a similar pattern was observed. All the examined ratios displayed a reduction at Day 21 by the addition of 10% *w/w* to OMSW recording an increase for both TSS and RSS while the substrate of 100% *w/w* of PS was found to be increased only for TSS. During the fermentation procedure, the fungus produces enzymes in order to degrade the lignocellulosic material. TSS's content reduction could be rationalized by the

fact the employed microorganism utilized them to obtain energy as well as to accomplish some other cellular activities. On the other hand, the observed TSS increase could be possibly attributed to the nature of the lignocellulosic substrate [29,42,44].

RSS increase may be due to the excess of enzymes secretion in order to participate in the degradation process. Furthermore, this trend for the depletion of RSS for almost all the examined ratios can be exemplified by the fact that *P. ostreatus* consumed fermentable sugars as an energy source for its growth.

Mushrooms are well known for their rich nutritional composition. They are composed of various bioactive substances such as *β*-glucans which promote human and animal health. *β*-Glucans form a major component of mushrooms' cell walls which are originated from their fruiting bodies [45]. Generally, *β*-glucans are *β*-d-glucose polysaccharides which are linked with 1,3-1,6 *β*-glycosidic bonds. Many previous studies have reported their antitumor and immune-stimulating properties [45–47]. In the present study, all examined substrates revealed a significant increase of the *β*-glucans content at the end of fermentation (Day 21). LPs addition by 20% *w/w* to PS recorded the highest value of *β*-glucans content whilst the highest increase was observed for the substrate of 100% *w/w* of PS. The increased *β*-glucans content to the fermented substrates is associated with *P. ostreatus* biochemical process who is utilizing substrates' nutrients, resulting thus to their increment. This procedure generates a novel fermentation outcome as a proteinaceous animal feed consisted of 1,3-1,6 *β*-glucans contributing to the economic and environmental impact in the context of the circular economy.

The final goal of our study was the development of novel feedstuffs with enriched nutritional composition by investigating the suitability of the aforementioned substrates. We have performed some primary experiments by incorporating the fermented substrates which have been already studied, in pigs' diet at the weaning stage of 30 to 75 days in a ratio of 5% *w/w* up to 10% *w/w*. The obtained results revealed their potential as feedstuffs, contributing to their health as well as in their body weight gain (data unpublished). Finally, it must be noted that all aforementioned raw materials were considered safe, since no microbial load of pathogenic microorganisms, aflatoxins or mycotoxins was detected.

## 5. Conclusions

In conclusion, this study herein, highlighted the importance of utilizing novel raw materials, like OMSW, PS and LP and their mixtures via SSF, by their bioconversion to proteinaceous animal feed contributing thus, to the reduction of the environmental impact. Results obtained herein revealed that all examined substrates which underwent SSF with *P. ostreatus* process, displayed a positive effect by their bioconversion into high added value products, enhancing thus their nutritional properties. This fact encourages their potential for use as animal feed. Regarding the results of our study, OMSW, PS, LP and their mixtures are strongly recommended to be used in the field of animal nutrition as proteinaceous dietary supplements to animals' nutrition. Their nutritional upgrade via SSF initiated by *P. ostreatus* indicates their potential as animal feed enhanced with bioactive compounds such as *β*-glucans which have beneficial impacts on animals' health and welfare. Furthermore, by using these substrates as feed supplements, animals can fulfill their nutritional needs, hence these upgraded substrates are nutritionally equal to commercial feedstuffs. Overall, the utilization of agro-industrial wastes contributes to the reduction of production cost, manages the disposal and pollution associated issues and finally, diminishes the environmental impact in the context of the circular economy.

**Author Contributions:** Conceptualization, C.E., S.A.H. and D.A.; methodology, C.E., G.M., S.A.H. and D.A; formal analysis, C.E. and D.A.; investigation, C.E., A.K. and D.A.; writing—original draft preparation, C.E; writing—review and editing, G.M., S.A.H. and D.A.; supervision, D.A. All authors have read and agreed to the published version of the manuscript.

**Funding:** This research has been co-financed by the European Union and Greek national funds through the Operational Program Competitiveness, Entrepreneurship and Innovation, under the call RESEARCH—CREATE—INNOVATE (Grant Number: T1EDK-04331).

**Institutional Review Board Statement:** Not applicable.

**Informed Consent Statement:** Not applicable.

**Data Availability Statement:** Not applicable.

**Conflicts of Interest:** The authors declare no conflict of interest.

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
