# Peer review of "Enrichment of Pistachio Shell with Olive Mill Waste or Lathyrus clymenum Pericarp Mixtures via Solid State Fermentation with Pleurotus ostreatus"

_fermentation, doi:10.3390/fermentation8020059_

Round 1

Reviewer 1 Report

please, find my comments attached

Reviewer 2 Report

Dear authors,

The revalorization of residues through solid state fermentation is very interesting. However, there is lack of results discussion in the paper. Some comments can you read below:

-In the abstract secton, a full stop is missing after P. ostreatus

- Line 107-109. Phrase needs revision

- Section 2.1 should be improved and more information should be included. Were the solid sterilized dried or with the wáter?, final moisture content?, was the size of the solid reduced by milling?, how was the fungus stored at 4 ºC?

- Section 2.2. More information should be included like the spore concentration of the inoculum, where were fermentations carried out?

- Line 130. After “dry weight” a full stop is missing

- Table 1. Could you explain why moisture content increase during the fermentation?

- Table 1: Explain TSS differences between time 0 and time 21

- Line 234. Is “For the 10%w/w addition..” correct?

- In line 285 it is mentioned that ratio of 60%  is considered as the most essential moisture content for fungal growth. However, optimum moisture content for growth depends on the organisms and substrate used for cultivation.

- Line 308-310: It is not clear. Do you mean the nutrients are transfered to the substrates?. Nutrients are contained in the substrates.

- Line 319-324. Please elaborate more this explanation is not clear.  

- Line 328. You should relate the fact that the content of cellulose in the fermented solid increased with the advantage of using this fermented solid for animal feed.

- Line 337-339. The increase of ash content after fermentation is not clear. Explain if the influence of ash content in the use of fermented solids for animal feed.

- Line 340-344. In all cases reducing sugars are not reduced after fermentation. How do you explain this effect?

- Line 353-357. It is not explained why b-glucans are increased in some cases after the fermentation.

- The paper does not compare with results of other published articles to valorate if they are in the same magnitud order.  

- Conclusion section is very extensive. This section should be modified as it seems more a discussion section

Reviewer 3 Report

The manuscript titled “Enrichment of olive mill waste mixtures with pistachio shell or Lathyrus clymenum pericarp via solid state fermentation with Pleurotus ostreatus” investigated solid state fermentation with the mixtures of Olive Mill Stone Waste (OMSW) with Pistachio Shell (PS) and PS with Lathyrus clymenum pericarp (LP) at various proportions and their nutritional composition was tested. This study provided possible strategy for the application of agro-industrial wastes as proteinaceous animal feed. However, there is still something is not clear and more data are suggested to be added to support the conclusion. The authors need pay attention to the following points.

  1. The composition of SSF substrates, including OMSW, PS and LP, should be tested in this study. It is important to illustrate why different fermentation results were obtained at various proportions.
  2. The details of proteins content determination should be provided, which is key parameter for the evaluation of fermented product.
  3. The vitro digestion study was suggested to be conducted to compare the fermentation product at various proportions. And thus, the application potential of these fermented products animal feed can be evaluated.

Round 2

Reviewer 2 Report

Dear author,

The paper has been improved after the revision. However, there are a few issues to be considered.

  • Still the following information is not clear "Solid materials size was found to be decreased compared to the initial, after milling process by using a filter with 2mm diameter which allowed oxygen diffusion"
  • In 2.1 section, you mentioned the percentage of inoculum to be added but not the concentration of cells in the inoculum
  • Line 336-337. The part "..which permits the fungus to transfer the nutrients such as nitrogen and iron, to a distance into the poor lignocellulosic substrates" is not clear. Do you mean that the fungus gets the nutrients required for its growth from the lignocellulose.
  • Line 348. "due to the different the fiber fractions". Delete the second "the".

Kind regards

Reviewer 3 Report

The manuscript is partly revised in light of some review comments. However, there was still some issue which was not clarified and no data to support the authors’ opinions

  1. Since the “protein content” was the most important parameter in this research, the crude protein content of SSF substrates, including OMSW, PS and LP, should be determined by AOAC’s official method. It is important for the substrate selection.
  2.  As answered by the authors, proteins’ determination was performed according to Kjeldahl method. So, the protein is not the true/real protein. The description “protein” should be replaced by “crude protein”.
